# Comparison of the Biological Basis for Non-HIV Transmission to HIV-Exposed Seronegative Individuals, Disease Non-Progression in HIV Long-Term Non-Progressors and Elite Controllers

**DOI:** 10.3390/v15061362

**Published:** 2023-06-13

**Authors:** Joseph Hokello, Priya Tyagi, Shelly Dimri, Adhikarimayum Lakhikumar Sharma, Mudit Tyagi

**Affiliations:** 1Department of Biology, Faculty of Science and Education, Busitema University, Tororo P.O. Box 236, Uganda; hokello.joseph@kiu.ac.ug; 2Cherry Hill East High School, 1750 Kresson Rd, Cherry Hill, NJ 08003, USA; pxt105@gmail.com; 3George C. Marshall High School, Fairfax County Public Schools, 7731 Leesburg Pike, Falls Church, VA 22043, USA; shelly.dimri@gmail.com; 4Center for Translational Medicine, Thomas Jefferson University, 1020 Locust Street, Philadelphia, PA 19107, USA; lakhikumarsharma.adhikarimayum@jefferson.edu

**Keywords:** HIV, genetic, seronegative, long-term non-progressors, elite controllers

## Abstract

HIV-exposed seronegative individuals (HESIs) are a small fraction of persons who are multiply exposed to human immunodeficiency virus (HIV), but do not exhibit serological or clinical evidence of HIV infection. In other words, they are groups of people maintaining an uninfected status for a long time, even after being exposed to HIV several times. The long-term non-progressors (LTNPs), on the other hand, are a group of HIV-infected individuals (approx. 5%) who remain clinically and immunologically stable for an extended number of years without combination antiretroviral therapy (cART). Meanwhile, elite controllers are comprise a much lower number (0.5%) of HIV-infected persons who spontaneously and durably control viremia to below levels of detection for at least 12 months, even when using the most sensitive assays, such as polymerase chain reaction (PCR) in the absence of cART. Despite the fact that there is no universal agreement regarding the mechanisms by which these groups of individuals are able to control HIV infection and/or disease progression, there is a general consensus that the mechanisms of protection are multifaceted and include genetic, immunological as well as viral factors. In this review, we analyze and compare the biological factors responsible for the control of HIV in these unique groups of individuals.

## 1. Introduction

Although heterosexual intercourse accounts for the vast majority of HIV-1 transmissions in Sub-Saharan Africa, exposure to human immunodeficiency virus (HIV) does not always result in infection. At present, there are many reports that indicate that there are groups of individuals who have been exposed to HIV on numerous occasions, but do not exhibit clinical or serological evidence of infection, suggesting mechanisms of natural resistance to HIV infection [1,2]. These people are referred to as HIV-exposed seronegative (HESN) individuals who are represented among HIV-serodiscordant couples and individuals with high-risk sexual behavior, such as commercial sex workers [3].

On the other hand, the vast majority of untreated HIV-infected individuals usually exhibit evidence of sustained viral replication marked by progressive CD4+ T-cell depletion [4,5,6,7,8,9,10,11]. However, there is a small percentage (5–10%) of individuals who remain clinically and immunologically stable for extended periods [12,13,14,15]. These individuals were initially referred to as “long-term non-progressors” (LTNPs) and were primarily classified based on the duration of infection and CD4+ T-cell count since viral load testing was not available until the mid-1990s. With the introduction of virologic testing, it was discovered that most LTNPs had low to moderate levels of viremia. Subsequent follow-up studies revealed that many LTNPs experienced progressively increasing viral loads and declining CD4+ T-cell counts [16,17,18,19]. The recognition of LTNPs’ ability to resist disease progression and remain free of symptoms for prolonged periods occurred approximately ten years into the HIV-1 pandemic. Initially, LTNPs were classified based on the absence of HIV-1-related disease manifestations. However, as viral load tests became accessible, the heterogeneity of these patients and their spontaneous control of HIV-1 replication became apparent. To date, two subgroups of HIV-1-positive patients with non-progressive disease courses have been identified. These include elite controllers (ECs) (also known as “elite suppressors”), who constitute a small fraction (about 0.3%) of untreated HIV-1-positive individuals [20,21] that spontaneously and durably control viremia and have undetectable viral loads when tested with the most sensitive commercial PCR assays. In contrast, conventional LTNPs are characterized by maintaining low-level viremia without treatment, typically having fewer than 2000 copies of viral RNA per milliliter of plasma. These individuals make up to 9% of all HIV-1-positive patients [22]. The duration of HIV-1 control in both EC and LTNPs can vary; however, numerous epidemiological studies have demonstrated that spontaneous control of HIV-1 replication can last for remarkably long periods, exceeding 30 years in some cases; although, some individuals may experience a loss of immune control [23,24]. The progression to acquired immunodeficiency syndrome (AIDS) in the presence of low or undetectable HIV-1 replications is a perplexing phenomenon observed in HIV-1 controllers, and its underlying causes remain unclear. There is no consensus, at present, regarding the factors that make HIV unfit in the cases of HESN, LTNPs, and EC. It is, however, well-known that these unique situations are certainly multifactorial and include contributions from the host, such as genetic, immunological, and behavioral features, as well as from the virus, such as viral tropism, load, and subtype. This review analyzes and compares the biological factors mediating non-HIV transmission in HESN individuals and non-disease progression in LTNPs and ECs.

## 2. Mechanisms of Non-HIV Transmission to HESN Individuals

### 2.1. Adaptive Immunity in HESN Individuals

#### 2.1.1. Humoral Immune Responses in HESN Individuals

Among the various ways of HIV transmission, sexual contact is the most common route. Globally, the heterosexual mode of viral spread is the major route of virus transmission. Initially, it was believed that the risk of heterosexual transmission was low as the virus was confined among men having sex with men (MSM) [25]. However, the transmission of HIV through heterosexual intercourse subsequently came to light. The initial aspect to consider during early mucosal transmission is the amount and the site of the virus being transmitted. It is presently known that HIV is present in female genital secretions [26] and the semen of HIV-infected men [27]. In this case, both cell-free and cell-associated viruses have been found in genital secretions, which can infect the epithelial cells lining the mucosa to transmit HIV during sexual intercourse [28]. It is well-documented that a cell-associated virus is more efficient than a cell-free virus in causing the mucosal infection of HIV [29,30,31,32]. In order to cross the epithelium, HIV triggers the degradation of epithelial integrity through a loss or disruption of tight junctions to allow the passage of the virus to gain access to susceptible cells of submucosa [33]. Transcytosis is another mechanism through which HIV can breach the epithelial barrier of the mucosa [34,35,36]. During transcytosis, viral particles, after binding to the surface molecules of the epithelial cell, such as heparan sulphate proteoglycans (HSPGs) or Galcer, gain access to the intracellular compartments of a cell [37,38,39]. Usually, the internalized virus in those cytoplasmic compartments is unable to complete its lifecycle. However, once trapped, the infectious virus is released into the external basal space, HIV infects the intraepithelial CD4+ cells or is transferred by dendritic cells (DCs) to other places. Accordingly, in human studies of vaginal HIV transmission, Shen et al. [40] demonstrated that DCs, macrophages and lymphocytes were the first cells to take up the virus following translocation through the pleuristratified epithelium of the lower female genital tract. However, subsequently, lymphocytes from both vaginal and ectocervix mucosa supported the highest levels of HIV-1 infection and replication [40,41].

Neutralizing antibodies (NAbs) inhibit HIV infection by binding to HIV envelope proteins, thereby prevents HIV interaction with the cellular receptors that leads to viral entry (Figure 1) [42,43,44,45,46]. Interestingly, Devito et al. [47] found that HIV-1 specific immunoglobulin A (IgA) isolated from the mucosa and plasma of HESN individuals effectively and potently inhibits HIV-1 transcytosis across the mucosal epithelial cells layer, thus demonstrating a crucial mechanism for HIV protection in HESN individuals. Similarly, Hocini and Bomsel [48] also observed that secretory IgA or IgG purified from colostrum obtained from HIV-seropositive women also impairs HIV transcytosis. 

#### 2.1.2. CD8+ T-Cell Responses in HESN Individuals

CD8+ T cells or cytotoxic T lymphocytes (CTLs) target and kill those cells that are infected with intracellular pathogens, including viruses. Intracellular pathogens, such as viruses, after being processed into small peptides, are presented along with Class I MHC molecules on the surface of antigen-presenting cells (APCs). These peptides are then recognized as foreign entities by CTLs and, thus, the infected cells are targeted and killed by CTLs in order to restrict the spread of the infection. Many studies, including a very recent one, have established a crucial role of HIV-specific CD8+ T cells in individuals with HESN [49]. 

Notably, Kaul et al. [50] demonstrated the presence of HIV-1-specific CD8+ T-cell responses in both the blood and cervix of HESN female sex workers in Nairobi, Kenya. On the contrary, they observed that these responses were not present in low-risk control groups. They observed that HIV-specific responses in the absence of detectable HIV infection in the genital mucosa of sex workers played an important role in protective immunity against heterosexual HIV-1 transmission. Furthermore, in 2002, Furci et al. [51] found that one of the mechanisms of resistance to HIV-1 infectivity in HESN individuals with natural protective immunity against HIV-1 was mediated by CD8+ T lymphocytes, which exerted a broad-spectrum non-cytotoxic HIV-1 suppressive activity, following cell-to-cell contact with the CD4+ T lymphocytes. In 2003, Truong et al. [52] showed another mechanism that restricted HIV-1 infection in HESN individuals. They reported that there was a resistance to peripheral blood mononuclear cell (PBMC) infection with HIV-1 in 21 (46.4%) of the 45 HESN individuals who were intravenous drug users in Vietnam. They further observed that PBMCs’ resistance to HIV infection was mediated either by an inhibition to HIV replication in CD4+ T lymphocytes or CD8+ T-lymphocyte-mediated virus suppression. Indeed, the observations of Furci et al. and Truong et al. are consistent with the report of Paxton et al. [53], who reported that CD8+ T lymphocytes from highly HESN individuals with natural protective immunity against HIV had greater anti-HIV activity than CD8+ T lymphocytes from non-exposed controls. Furthermore, Paxton et al. reported that purified CD4+ T lymphocytes from HESN individuals with protective innate immunity against HIV were resistant to infection with multiple primary isolates of HIV-1 than CD4+ T lymphocytes from non-exposed controls. Consistent with this observation, in a separate set of experiments, Paxton et al. [54] subsequently demonstrated that the resistance to HIV-1 infectivity of CD4+ T cells from HESN individuals was due to the low expression of the chemokine receptor, CCR5 HIV co-receptor and high production of β-chemokines, including MIP-1α, MIP-1β and RANTES, which are the natural ligands to CCR5. Notably, in addition to CD4+ T cells, HIV also infect macrophages, which also express the HIV receptor and coreceptor. Interestingly, it was found that macrophages derived from HESN individuals were similarly unsusceptible to HIV infection, similar to the CD4+ T cells [55]. Consequently, it is imperative to determine whether or not this observed resistance to HIV infection in HESN individuals can be genetically inherited by their offspring, which would have implications for mother-to-child HIV transmission. 

### 2.2. Innate Intracellular Antiviral Proteins in HESN Individuals

HIV-1 interacts with many host cell proteins in order to complete its lifecycle and generate new progeny [56,57,58]. Some cellular proteins are required for HIV-1 replication, while others are negative and impede the HIV lifecycle; thus, they need to be countered. For instance, the cellular co-factor, known as Lens epithelium-derived growth factor p75 (LEDGF/p75) [59], plays a role in promoting the integration of viral cDNA (provirus) into the host cell genome [60,61]. On the other hand, there are negative cellular factors, such as tri-partite motif 5alpha (TRIM5α) [62], tetherin (BST2, CD317) [63] and apolipoprotein B mRNA-editing catalytic polypeptide-like 3G (APOBEC3G, A3G) [64], which inhibit the HIV lifecycle through distinct antiviral activities. These cellular factors exert antiviral mechanisms through different mechanisms, such as TRIM5α disrupting the structure of the retroviral capsid, thereby interfering with the natural ‘uncoating’ process in a species-specific manner [65,66]. Additionally, TRIM5α has recently been found to promote innate immune signaling, whereas Tetherin inhibits HIV by restricting the release of newly formed viral particles from the cell membrane of the producing cell [63]. APOBEC3G inhibits HIV replication by inducing G-to-A hypermutation in the HIV-1 genome [62,63,64,67].

Previously, Speelman et al. [68] observed that impaired HIV entry alone was not sufficient to explain the reduced CD4+ T-cell susceptibility to HIV-1 infection among HESN individuals. They further observed that, in addition to HIV-1 entry inhibition, additional cellular restrictions of HIV-1 might account for the continued seronegativity of HESN individuals. Indeed, Gonzalez et al. [69] investigated the expression levels of antiviral factors, including APOBEC3G, TRIM5α, RNaseL, and SerpinA1, in the mucosa of HESN individuals. They found that HESN individuals exhibited higher levels of expression of all antiviral factors in the genital mucosa, and oral and peripheral blood mononuclear cells (PBMCs) compared to the healthy controls. This suggests that the antiviral activities of these factors are compartmentalized such that these proteins are accumulated at the HIV-1 exposure sites. Thus, these proteins have predominant roles, depending on the tissue to prevent infection, which results in reduced viral loads and modulated susceptibility to HIV infection. Similarly, Mous et al. [70] analyzed the expressions of LEDGF/p75, APOBEC3G, TRIM5α and Tetherin in a Senegalese cohort of HESN individuals in order to determine whether these antiviral proteins possibly contribute to resistance to HIV infection. Contrary to the observation of Gonzalez et al., they reported no significant differences in the expression levels of APOBEC3G, TRIM5α and Tetherin in the PBMCs of HESN individuals. However, they observed a marked difference in the expression levels of LEDGF/p75 (lower level in LEDGF/p75 expression in HESN), suggesting that LEDGF/p75 plays an important role in mediating resistance to HIV-1 infection by HESN individuals. 

Overall, it was noted that relatively lower inflammation in the system is the main protection factor, as investigators found that HESNs individuals maintained a low-inflammatory “immune” profile in their blood and genital mucosa [71,72,73]. In fact, the research indicates that the vaginal mucosa of HESN commercial sex workers has high amounts of neutralizing and anti-inflammatory proteins, such as anti-proteases [72,73,74].

Lower levels of pro-inflammatory cytokines, including TNF-α, IL-1 and IFN-γ, as well as monokine induced by IFN-γ (MIG) and IFN-γ-induced protein (IP)-10 chemokines, have been documented in cervicovaginal lavages of HIV-exposed seronegative (HESN) individuals, when compared to HIV-infected commercial sex workers [75,76]. It is noteworthy that the production of MIG and IP-10, which is induced by the expression of IFN-γ, and certain polymorphisms in the interferon regulatory factor 1 (IRF-1) that regulate IFN-γ have been associated with protection against HIV infection [77,78]. In a Kenyan female commercial sex workers cohort, varying states of cell stimulation and cytokine release among HIV-specific CD4+ and CD8+ T-cell populations resided in both the blood and genital tract [50,79,80,81]. In these individuals, a low-activation T-cell profile correlated with a greater ability to proliferate in response to HIV p24 peptides, when compared to that observed in HIV-infected persons. Similarly, Ritchie et al. [82] compared the HIV-specific T-cell responses between HESN individuals and their unexposed counterparts. They observed that, although there was a low frequency of HIV-specific T cells in both groups, over time, the response rates and the magnitude of HIV-specific responses were significantly higher among HESN individuals mediated by CD4+ T cells. Consistent with this observation, Murashev et al. [83] reported that there was multiple HIV-specific cytokines production by both CD4+ and CD8+ T cells in the vast majority of HESN individuals.

### 2.3. Innate Immunity in HESN Individuals

Human innate marginal zone (MZ)-type B-cells serve as an early first line of adaptive defense against invading pathogens because of their ample presence in the lymphoid organs and mucosal-associated structures [84]. MZ B-cells can also traffic antigens to follicular B-cell areas of lymphoid structures and promote T-cell-dependent B-cell class switch gene recombinations and affinity maturation resulting in the production of many specific antibodies with potent neutralizing and antibody-dependent cell cytotoxicity (ADCC) effector functions [85]. Notably, through its surface lectins, MZ B-cells have been shown to interact with the fully glycosylated gp120 of HIV. Moreover, this interaction in the presence of the blood B-lymphocyte stimulator (BLyS/BAFF) has been documented to result in the production of poly-reactive IgA and IgG antibodies, which recognizes gp120 [86].

Some reports demonstrated that, during the course of HIV infection, the expression of BLyS/BAFF is increased. However, the level is never restored following antiretroviral therapy [87,88]. Soluble HIV-Nef is reported to directly modulate BLyS/BAFF membrane expression and soluble release by derived monocyte-derived DCs [89]. HIV-Env also up-regulate BLyS/BAFF expression in macrophages and its release [86]. Similarly, BLyS/BAFF has been demonstrated to be directly induced by type-I IFNs [90,91]. 

Microbial translocation is also shown to up-regulate BLyS/BAFF expression and its secretion [89,90,91,92]. Multiple studies have provided evidence that BLyS/BAFF overexpression in the bloodstream of HIV-infected progressors is associated with hyperglobulinemia and elevated frequencies of precursor-like MZ B-cells expressing IL-10 [87,88,89,90,91,92,93]. However, in contrast to these findings, both the levels of BLyS/BAFF in the blood and the frequencies of precursor-like MZ B-cells remained unchanged in HIV-infected elite controllers [87,93]. 

Similarly, in the Benin cohort, plasma and cellular BLyS/BAFF levels were significantly lower in the blood of HESN commercial sex workers when compared to HIV-infected commercial sex workers and HIV-uninfected non-sex workers [94]. The low BlyS/BAFF levels in these HESN commercial sex workers correlated with the low-inflammatory response previously described in these individuals [76], suggesting that they could also contribute to the maintenance of low-inflammatory conditions observed in HESN individuals. Based on these observations, the capacity to control BLyS/BAFF expression in HESN individuals seems to coincide with natural immunity against HIV, whereas the excessive expression of BLyS/BAFF may promote immune dysregulation [95]. On the other hand, other studies have demonstrated that the protection against HIV infection observed in HESN individuals was associated with heterozygosity for delat32 polymorphism of the CCR5 chemokine receptor gene [96,97]. Most recently, Munusamy et al. [98] investigated the role of mucosal immune responses and cervicovaginal microbiomes in preventing HIV infection in HESN women. They observed that HESN females’ genital tracts are a microenvironment well-equipped with innate immune factors, including NK cells, antiviral cytokine mediators that offer protection against HIV acquisition. The observations of Ponuan et al. (2021) are consistent with the findings of Tomescu et al. [99], who reported increased NK and dendritic cell activities in HESN individuals. Saulle et al. [100] extensively investigated the immunological parameters associated with resistance to HIV infection in HESN individuals and found out that there was increased immune activation in HESN individuals, which, however, was not mediated by gastrointestinal barrier and microbial translocation. This is consistent with the report of Fenizia et al. [101] who observed that natural protection against HIV acquisition in HESN individuals is associated with immune activation, with the increased expression of immune activation markers at both basal state and following specific stimulation. Immune activation state in HESN individuals can influence the induction of specific adaptive antiviral immune responses. 

## 3. Mechanisms of Disease Non-Progression in HIV LTNPs

### 3.1. Attenuated Viruses in LTNPs

The reduced fitness of HIV-1 strains is associated with their inability to evade the host immune system. Numerous studies have consistently demonstrated that virus strains isolated from long-term non-progressors (LTNPs) exhibit lower fitness compared to strains from progressors, making them less capable of evading the host’s immunological response [102,103,104,105]. The Sydney Blood Bank Cohort is an example of typical HIV-1 LTNP viral attenuation. In this particular instance, individuals became infected through blood transfusions from a single donor who carried an HIV-1 strain with a significant deletion in both the long-terminal repeat (LTR) and Nef gene [106,107]. Remarkably, six recipients of the blood cohort subsequently developed a long-term non-progressor (LTNP) status, demonstrating stable CD4+ T-cell levels and maintaining low viral loads for a duration exceeding 10 years. Studies were also conducted to compare the replication efficiency of different viral strains from HIV progressors and LTNPs. The finding suggested that the viral strains from LTNP replicate at a much slower rate and are less infective [108]. In this study, Choudhary et al. [108] found that, in the absence of interleukin-10 (IL-10) and transforming growth factor beta (TGF-β), which upregulate CCR5 viral co-receptor expression, critical factors determine the reduced pathogenesis of HIV-1 in LTNPs. Consequently, highly reduced infectivity and less cytopathic effects were observed in LTNP than in HIV-1 progressors. These observations suggest that LTNPs harbor HIV strains with specific mutations, which significantly attenuate HIV fitness and reduce infectivity [104,105]. Nevertheless, reduced viral fitness, similar to the mentioned case, is unlikely to entirely account for the ability of long-term non-progressors (LTNPs) to control viral replication. This is particularly evident when considering other LTNPs who harbor replication-competent viruses [109,110].

### 3.2. Innate Intracellular Antiviral Proteins in LTNPs

#### 3.2.1. Endogenous Antiretroviral Protein APOBEC3G

APOBEC3G, a cytidine deaminase, causes excessive G to A hypermutations in viral DNA [111]. Lower levels of APOBEC3G mRNA in HIV-infected persons are associated with an increased risk of progression to AIDS, while high levels of APOBEC3G mRNA causes a slow HIV disease progression [112]. These findings were also recapitulated by Vazquez-Perez et al. [113], where they documented a direct correlation of APOBEC3G mRNA expression levels with higher CD4+ T-cell counts and lower HIV-1 viral loads in the PBMCs from a group of HIV-1-infected individuals. Furthermore, another study found that treatment-naive persons with lower viral loads and HIV infection of longer than 3 years had higher levels of APOBEC3G expression than both progressors and uninfected healthy controls. In a large cohort study of HIV-infected Kenyan women, APOBEC3G-induced hypermutations in env/vpu proviral DNA correlated with higher CD4+ T-cell counts [114]. In the year 2000, An et al. demonstrated that APOBEC3G polymorphisms are associated with increased rates of disease progression [115]. In this study involving 3073 HIV patients, it was found that the presence of the APOBEC3G exon 4 variant allele H186R was associated with accelerated progression to AIDS and a notable decline in CD4+ T-cell counts, when compared to individuals who did not possess this particular polymorphism. Given these observations, it is highly conceivable that the high levels of attenuated viruses observed in LTNPs are a direct result of the hypermutation effects of APOBEC3G. 

#### 3.2.2. The Trim5α

Trim5α plays a crucial role in targeting HIV-1 during the post-cell-entry phase by binding to the HIV-1 capsid protein and inhibiting viral uncoating [116]. Elevated levels of Trim5α have been linked to slower HIV-1 disease progression, and the R136Q polymorphism is believed to be particularly effective in blocking X4-tropic HIV-1 infection [116]. In 2006, Javanbakht et al. [117] investigated the impact of 12 common Trim5α single nucleotide polymorphisms (SNPs), including R136Q, in 939 HIV-1 seroconverters from five HIV-1 natural history cohorts in the USA. They found no significant association between these SNPs and disease progression to AIDS or low CD4+ T-cell counts. In a separate study, Nissen et al. [118] conducted the whole-exome sequencing of HIV-1 long-term non-progressors and identified rare variants in genes encoding innate immune sensors and signaling molecules.

### 3.3. The Role of Genetic Factors in Disease Non-Progression in HIV LTNPs

Human leukocyte antigen (HLA) class-I alleles A, B and C present digested components of an internalized cellular pathogen as antigens to cytotoxic CD8+ T lymphocytes. Heterozygous HLA class-I genotype has been linked to slower HIV-1 disease development and reduced viral load by allowing the presentation of a broader spectrum of peptides of HIV-1, which lowers the likelihood of viral-escape variations [111]. HLA-B57 variations, in particular, B5701 and B5703, control HIV-1 infection by eliciting a cross-reactive response against immunodominant Gag epitopes [111,119]. HLA-B27, on the other hand, is a well-studied allele in terms of HIV-1 disease non-progression, but was found not to be as statistically significant in controlling HIV disease progression in large cohorts [119,120]. Approximately 10% of Caucasians, Africans and Asians carry a copy of the HLA-B57 allele and 8% of the persons in major continental populations carry the B27 allele [121]. One important factor in the consideration of the significance of B57 and B27 alleles in regard to HIV disease progression is the viral load parameters applied to different LTNP cohorts. Research studies that follow the most stringent criteria for defining long-term non-progressors (LTNPs), such as a viral load of 75 RNA copies per milliliter of plasma, often observe a high prevalence of individuals who test positive for B57 but negative for B27 [122]. There are significant differences in B57 and B27 positivity when comparing LTNP to the HIV progressor. The study indicated that B57, rather than B27, was found at a considerably higher frequency in LTNPs than in both progressors and healthy seronegative controls. Other HLA class-I alleles reported to mediate a slow rate of HIV disease progression include B13, B15, B44, B51 and B58, and in some cohorts, as many as 90–95% of LTNPs carry at least one of these alleles [122]. On the other hand, Walli et al. [123] demonstrated that LTNPs who were heterozygous for the delta32 CCR5 chemokine receptor gene exhibited significantly slower plasma viral loads than the normal progressors. This observation strengthens the hypothesis of a favorable influence of the CCR5 delta32 genotype on the progression of HIV disease. 

### 3.4. The Role of Adaptive Immune Responses in LTNPs

#### 3.4.1. Humoral Immunity

The presence of broad acting neutralizing antibodies (NAbs) is not common in HIV-1-infected people, with contradictory data on their prevalence in LTNPs. Notably, a number of research groups noted the higher prevalence of NAbs in LTNPs than in progressors [124,125,126]; however, others have shown no significant difference in the levels of NABs between LTNPs and progressors [127,128]. Although, with small sample sizes, several studies have demonstrated that LTNPs have a stronger virus-specific Nab response than progressors against both autologous and heterologous viral strains [124,126,129,130,131]. However, it is interesting to note that, because of the limited viral replication and more varied virus populations, NAb levels in LTNPs peak late during infection. Progressors, on the other hand, rapidly lose their NAb response during the course of their infection due to a rapid decline in CD4+ T-cell density [124,132]. 

#### 3.4.2. CD8+ T Cells 

Cytotoxic T lymphocytes (CTLs), also known as CD8+ T cells, play a critical role in eliminating HIV-infected host cells and are considered the primary drivers of viral load reduction in HIV-1 controllers [133]. In the case of many long-term non-progressors (LTNPs), their CD8+ T-cell responses have been found to exhibit a high precursor frequency of CTLs. This results in the generation of broadly reactive CTLs that target conserved sequences within the *gag*, *pol* and *env* genes [133,134,135]. Ferre et al. [136] demonstrated that polyfunctional HIV-specific CD8+ T cells reacting to Gag p24 were more prevalent in HIV controllers’ rectal mucosa than in non-controllers or cART-treated individuals. Similarly, CD8+ T-cell responses against conserved areas of Gag, including HLA-B27 and -B57-restricted epitopes, were also observed in these patients’ sera. However, the caveat that applies to LTNP-CD4+ T cells also applies to their CD8+ T cells; that is to say, it is not known whether the high levels of virus-specific CD8+ T cells are the cause or effect of their low viral loads. 

## 4. Mechanisms of Spontaneous and Durable HIV Control by Elite Controllers

### 4.1. Adaptive Immune Response in HIV Elite Controllers

#### 4.1.1. Humoral Immune Responses 

The ability of elite controllers (ECs) to spontaneously and sustainably control HIV replication and disease course without antiretroviral therapy is incredible, and understanding the exact mechanisms of natural HIV control is critical for the development of immunotherapies against HIV. Recent reports have associated HIV control by ECs to adaptive immune responses following infection. Moris et al. [137] observed that ECs maintain large HIV-specific memory B-cell pools sustained by follicular helper T cells. They further observed that, although ECs rarely show high titers of neutralizing antibodies, the antibodies, when present, are capable of potently neutralizing contemporaneous viral strains and that the ECs uniquely display antibodies with polyfunctional effector functions. 

Accordingly, Lambotte et al. [138] reported significantly higher levels of antibody- dependent cellular cytotoxicity (ADCC) antibodies in ECs compared to viremic individuals. The ADCC responses correlated with strong CD8+ T-cell viral suppressive activity are presented in Figure 2. Previously, Lambotte et al. [139] demonstrated a heterogonous neutralizing antibody and ADCC responses in HIV ECs compared to viremic individuals. Most recently, Madhavi et al. [140] observed that ADCC in HIV ECs was associated with HIV Env- and Vpu-specific antibodies. 

Human endogenous retroviruses (HERVs) constitute about 8% of the human genome. Whereas the majority of HERVs are transcriptionally silenced, HERV-K, the most recently integrated HERV, remains transcriptionally active and specific HERV-K mRNA and proteins can be detected. Interestingly, de Mulder et al. [141] investigated antibody responses against HERV-K in relation to HIV infection in ECs. De Mulder et al. observed that HIV ECs had strong cellular and antibody responses against HERV-K Gag proteins, suggesting that anti-HERV-K antibodies targeting Gag proteins are involved in HIV control in ECs. 

#### 4.1.2. CD4+ T-Cell Responses

Virus-specific responses to CD4+ T lymphocytes are usually impaired in the early stages of HIV disease due to the considerable depletion of CD4+ T lymphocytes. The lack of HIV-specific proliferative CD4+ T-cell responses that are a blueprint for progressive HIV infection [142,143,144] were subsequently shown to be largely caused by a loss of functions, in particular, defective IL-2 secretion; however, not physical deletion [145,146,147]. In HIV-infected individuals, CD4+ T-cell responses are low and manyfold reduced than CTL responses [145]. HIV-specific CD4+ T cells maintain a proliferative capacity in vitro in ECs. This indicates that these responses may differ from those of individuals without treatment with progressive disease [148,149,150]. Consistent with these observations, HIV-specific CD4+ T cells of ECs produced more IL-2 and were more polyfunctional than those in disease progressors [128,145,147,151]. Follow-up research contrasting ECs, disease progressors and those with an effectively managed viral load on antiretroviral therapy showed that the variations in proliferative potential, IL-2 secretion and phenotypic markers observed were an outcome, and not the cause, of a low viral load. [151,152,153,154]. On the other hand, Chen et al. [155] investigated the HIV infectivity of CD4+ T cells from ECs compared to disease progressors and HIV-uninfected persons. They reported that, compared to the controls, the CD4+ T cells from ECs were less susceptible to HIV infection. They further observed that this resistance to HIV infectivity by CD4+ T cells of ECs was due to less-efficient reverse transcription and mRNA expression from proviral cDNA, which in turn was associated with the strong and selective up-regulation of the cyclin-dependent kinase (CDK) inhibitor p21. This observation suggests that CDK inhibitor p21 acts as a barrier to the HIV infection of CD4+ T cells from ECs by inhibiting CDK9, a cellular co-factor for HIV viral Tat protein required for the efficient expression of HIV genes.

#### 4.1.3. CD8+ T-Cell Responses in HIV Controllers

Several recent studies have identified CD8+ T-cell responses to be strongly associated with spontaneous and durable HIV control in ECs [156,157,158,159,160]. Compelling evidence supporting the pivotal role of CD8+ T-cell responses in the natural control of HIV comes from the association of specific HLA class-I B27 and B57 alleles with elite controllers (ECs). Numerous studies have consistently demonstrated this association [161,162,163,164]. HLA class-I alleles govern the presentation and recognition of HIV epitopes by the host immune response, and the presence of these protective HLA class-I alleles facilitates the development of robust and efficient CD8+ T-cell responses [165]. On the other hand, Monel et al. [166] reported that CD8+ T cells from ECs were able to recognize non-activated CD4+ T cells in the absence of de novo HIV protein synthesis and destroy such cells. Similarly, Buckheit et al. [167] demonstrated that CD8+ T cells in ECs were able to efficiently recognize and eliminate both resting and activated CD4+ T cells rapidly and prior to productive infection by HIV. Using full-length individual proviral sequencing, Jiang et al. (2020) determined a proviral reservoir landscape in ECs. They reported that, overall, the number of intact and defective proviruses were significantly lower in ECs compared to ART-treated individuals. They, however, noted that intact proviral sequences constituted a significant proportion of all proviral sequences in ECs [168]. Similarly, Lian et al. (2021) simultaneously analyzed individual proviral sequences, as well as their respective chromosomal locations, in order to produce a holistic picture of the proviral reservoir landscape in ECs [169]. They reported that there were unique profiles of intact reservoir proviruses in ECs that exhibited no evidence of CTL- or antibody-mediated escape mutations. However, these intact proviruses were instead integrated within heterochromatin regions of host DNA. The intact proviruses were characteristically different from the autologous defective proviruses from ECs, as well as the proviruses from individuals on prolonged ART, which were mostly integrated within euchromatin DNA regions [169]. The authors attributed the presence of the defective provirus profiles integrated within euchromatin regions to immense pressure from the host immune system leaving the intact proviruses integrated within heterochromatin regions, which, in turn, offers protection from being targeted by the host immune system as a result of an inability to become reactivated. Conceivably, these observations explain the observed reduced seeding of the latent HIV reservoirs in ECs, as well as the sustained low levels of viremia [170,171].

Accordingly, Julg et al. [170] found that recovery of HIV from ECs was difficult because of the low viremia levels; however, once recovered, the virus was fully infectious and replication competent in CD4+ T cells obtained from healthy individuals.

## 5. Conclusions

The analysis of the biological factors that mediate non-HIV transmission to HESN individuals and HIV non-disease progression in LTNPs and ECs reveal that the factors are obviously multifaceted and indeed convoluted. HESN individuals, several studies have documented the role of mucosal humoral and cellular-mediated immune responses to mediate the observed control in HIV transmission. Specifically, mucosal antibody and CD8+ T-cell responses in HESN individuals are able to effectively block heterosexual HIV infection. However, these responses are also accompanied by the inability of CD4+ T cells to support productive HIV infection and replication due to the low expression of CCR5, an HIV co-receptor that is utilized by viruses that constitute the majority of new HIV infections. Furthermore, the presence of innate immunity, such as MZ-type B-cells along with a low expression of BLyS/BAFF in HESN individuals, strongly correlated with low inflammatory reactions and protection against HIV transmission. Although the role of innate antiviral proteins, such as APOBEC3G, have been suggested as a potential mechanism in HESN individuals, it remains controversial. It is, however, imperative to note here that the natural protective immunity in HESN individuals already exists, prior to HIV exposure. Accordingly, HESN individuals represent a unique population and model for HIV preventive vaccines. 

In the case of HIV LTNPs, multiple studies have documented the fact that these group of individuals harbor HIV strains with low replicative fitness. Indeed, several studies that have sequenced HIV strains obtained from LTNPs observed that they harbor large-scale deletions in LTR and the *nef* gene. LTR is the viral promoter that regulates the expression of HIV genes, while the Nef protein is vital for the modulation of viral infectivity. Mutations in LTR and the *nef* gene thus indicate that the viral strains in the LTNPs are defective, with low viral fitness and a low capacity to replicate, in order to cause severe disease. However, the key question to answer is whether all LTNPs are initially infected by a defective virus, or whether HIV is restricted due to the antiviral activity of APOBEC3G, which has also been demonstrated to be highly expressed in LTNPs. On the other hand, the role of genetic factors, particularly the possession of HLA class-I B57 alleles along with efficient CD8+ T-cell responses, is also documented to play a considerable role in the control of HIV disease progression in LTNPs, which is similar to what is observed in ECs.

Lastly, ECs represent a unique model for immunotherapy and/or functional cure for HIV infection, as they are able to develop HIV-specific adaptive immunity that is capable of persistently suppressing viremia following infection. In this case, there is overwhelming evidence suggesting that ECs generate antibodies with an enhanced ability to mediate ADCC, which contributes to viral containment. Mounting evidence also supports the observation that potent CD8+ T-cell responses control HIV replication in ECs. Protective CD8+ T-cell responses were mediated by efficient antigen presentation by HLA class-I alleles, particularly B57, which are demonstrated to be prevalent in ECs, just as it is the case for LTNPs. It is imperative to note here that ECs are a subset of LTNPs, both of which harbor defective provirus profiles conceivably mediated by similar immune mechanisms. These observations suggest that therapeutic vaccines that are capable of generating polyfunctional antibodies and induce CD8+ T-cell responses would be able to achieve the durable control of viral replication or a functional HIV cure.

## Figures and Tables

**Figure 1 viruses-15-01362-f001:**
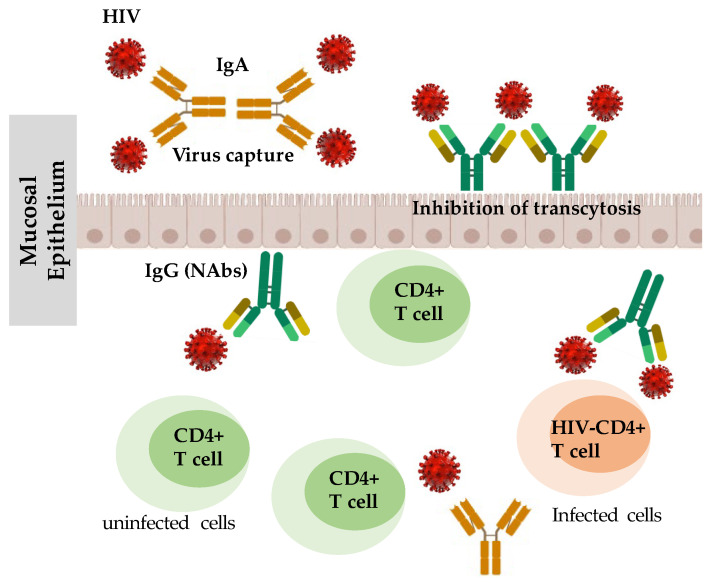
Neutralizing antibodies prevent HIV interaction with cellular receptors that hinders productive HIV infection. Neutralizing antibodies (e.g., IgA from mucosa and plasma or IgA and IgG from colostrum) prevent HIV by impairing HIV transcytosis.

**Figure 2 viruses-15-01362-f002:**
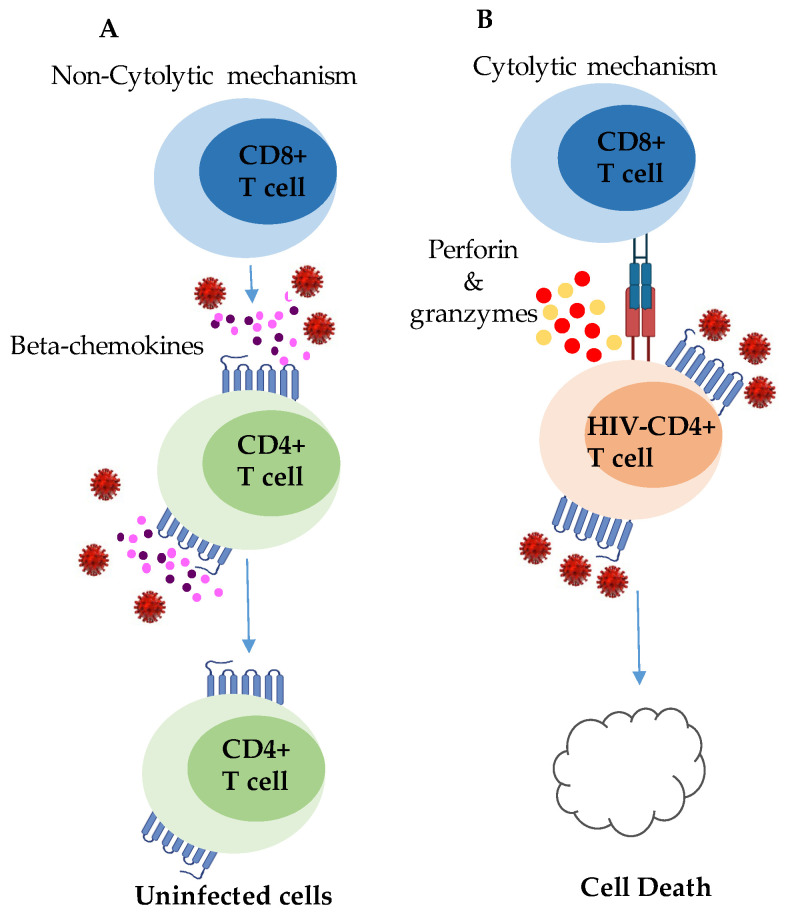
Contribution of CD8+ T cells in preventing HIV infection and replication. (**A**) Non-cytolytic functions of CD8+ T cells in HIV infection: in response to HIV infection, CD8+ T cells release beta-chemokines that act as protective factors for CD4+ T cells, preventing their infection. (**B**) Cytolytic functions of CD8+ T cells in HIV infection: when interacting with a CD4+ T cell that is actively producing HIV, CD8+ T cells employ cytolytic mechanisms by releasing perforin and granzyme. This leads to the destruction of the infected cell through the induction of cell death.

## Data Availability

Not applicable.

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
