# Peer review of "Comparison of the Biological Basis for Non-HIV Transmission to HIV-Exposed Seronegative Individuals, Disease Non-Progression in HIV Long-Term Non-Progressors and Elite Controllers"

_viruses, 2023, doi:10.3390/v15061362_

Round 1

Reviewer 1 Report

The authors conducted a review compared the biological basis for non-HIV Transmission among HIV-exposed seronegative individuals, disease non-progression in HIV long-term non-progressors, and elite Controllers. Overall, this paper is well written. After careful consideration, I am happy to suggest accepting this paper with minor revision.

Some punctuation errors need to correct before publication。 Such as in line 217, 227, 294, 344, etc.

Author Response

Following are point-by-point response to reviewers’ comment

Reviewer 1:

Comments and Suggestions for Authors

Comment 1

The authors conducted a review compared the biological basis for non-HIV Transmission among HIV-exposed seronegative individuals, disease non-progression in HIV long-term non-progressors, and elite Controllers. Overall, this paper is well written. After careful consideration, I am happy to suggest accepting this paper with minor revision.

Some punctuation errors need to correct before publication。 Such as in line 217, 227, 294, 344, etc.

Response

We sincerely thank the reviewer for the suggestion. Following the suggestion, we have made necessary amendment in line 217, 227, 294 and 344.

Thank you

Reviewer 2 Report

Overall, all the virological and immunological aspects related to HESN, LTNP and EC phenotypes are addressed in this review. Some aspects may be under cited. Some reference in these topics has been indicated.

The term HAART is everyday less used. It may be substituted by combination antiretroviral therapy (cART).

Reference 20 maybe accompanied or substitute by reference PMID: 16142675. This reference maybe the first in defining HIV-controllers.

Reference 22 is related to the loss of spontaneous viral control more than loss of immune control. Authors may change “immune control” by “viral control”, and add reference PMID: 29212942, which is seminal in relation to loss of viral control.

Line 97, the word “HIV” maybe suppressed.

The section “2.1.1 Humoral immune responses in HESN Individuals” is under cited. Most of the content of this section is related to mucosal HIV transmission, however only one citation is referred to humoral response in HESN individuals. Authors may cite more references related to this topic (E.g.: PMID: 22948273, PMID: 23945502, PMID: 25275513, PMID: 31173604, PMID: 33315937, among others).

Lines 206-214 are more related to HIV-specific T-cell response in HESN and no related to the title of 2.2 Innate intracellular antiviral proteins in HESN individuals. This information may be moved to the part of the text related to T-cell immunity in HESN.

Is reference 100 referred to LTNPs? Or are these individuals ECs?

The sentence in lines 278 – 280 seems incomplete.

In relation to low viral fitness in LTNPs authors may also refer to recent papers: PMID: 29636433, PMID: 35401460, among others.

Lines 357-367 is referred to adaptive immunity related to T-cell response, maybe a 3.4.2 section would be more appropriate.

Sentence in lines 439 – 441 is not accurate, e.g.: studies cited above associate attenuated strains of HIV in EC (PMID: 29636433, PMID: 35401460).

In the EC part, recent findings about the nature of the quality of HIV reservoir in EC is missing. Authors may include this information. PMID: 34910552, PMID: 32848246. Also, important will be to discuss this information in the context of the LTNP phenotype.

As a general observation, EC is a subgroup of LTNP, with the extreme control of viremia. Authors may clarify this in the conclusion as a cause of the overlapping mechanism involved in these phenotypes.

Author Response

Following are point-by-point response to reviewers’ comment

Reviewer 2:

Comments and Suggestions for Authors

Comments 1

Overall, all the virological and immunological aspects related to HESN, LTNP and EC phenotypes are addressed in this review. Some aspects may be under cited. Some reference in these topics has been indicated.

The term HAART is everyday less used. It may be substituted by combination antiretroviral therapy (cART).

Response

As advised, we have substituted HAART by combination antiretroviral therapy (cART).

Comments 2

Reference 20 maybe accompanied or substitute by reference PMID: 16142675. This reference maybe the first in defining HIV-controllers.

Response

We have added the reference as suggested

Comment 3

Reference 22 is related to the loss of spontaneous viral control more than loss of immune control. Authors may change “immune control” by “viral control”, and add reference PMID: 29212942, which is seminal in relation to loss of viral control.

Response

We have made necessary changes.

Comment 4

Line 97, the word “HIV” maybe suppressed.

Response

As suggested, we have the amendment.

Comment 5

The section “2.1.1 Humoral immune responses in HESN Individuals” is under cited. Most of the content of this section is related to mucosal HIV transmission, however only one citation is referred to humoral response in HESN individuals. Authors may cite more references related to this topic (E.g.: PMID: 22948273, PMID: 23945502, PMID: 25275513, PMID: 31173604, PMID: 33315937, among others).

Response

As suggested, we have updated the references

Comment 6

Lines 206-214 are more related to HIV-specific T-cell response in HESN and no related to the title of 2.2 Innate intracellular antiviral proteins in HESN individuals. This information may be moved to the part of the text related to T-cell immunity in HESN.

Response

As advised, the text has been updated accordingly

Comment 7

Is reference 100 referred to LTNPs? Or are these individuals ECs?

Response

In that reference LTNPs are referred as “Elite suppressors”, but reference is pertaining to LTNPs.

Comment 8

The sentence in lines 278 – 280 seems incomplete.

Response

Text is revised for better sense.

Comment 9

In relation to low viral fitness in LTNPs authors may also refer to recent papers: PMID: 29636433, PMID: 35401460, among others.

Response

Thanks for the suggestion; we have incorporated the suggested references.

Comment 10

Lines 357-367 is referred to adaptive immunity related to T-cell response, maybe a 3.4.2 section would be more appropriate.

Response

The Lines 357-367 deals with humoral immunity, specifically bNAbs/NAbs responses.

Comment 11

Sentence in lines 439 – 441 is not accurate, e.g.: studies cited above associate attenuated strains of HIV in EC (PMID: 29636433, PMID: 35401460).

Response

We have updated the references as suggested

Comment 12

In the EC part, recent findings about the nature of the quality of HIV reservoir in EC is missing. Authors may include this information. PMID: 34910552, PMID: 32848246. Also, important will be to discuss this information in the context of the LTNP phenotype.

Response

We have updated the references as suggested

Comment 13

As a general observation, EC is a subgroup of LTNP, with the extreme control of viremia. Authors may clarify this in the conclusion as a cause of the overlapping mechanism involved in these phenotypes.

Response

We appreciate reviewer comment. As suggested, we have clarified this in the conclusion section.

Thank you
